# Status Quo and Future Perspectives of Molecular and Genomic Studies on the Genus *Biomphalaria*—The Intermediate Snail Host of *Schistosoma mansoni*

**DOI:** 10.3390/ijms24054895

**Published:** 2023-03-03

**Authors:** Ming Fung Franco Au, Gray A. Williams, Jerome H. L. Hui

**Affiliations:** 1School of Life Sciences, Simon F.S. Li Marine Science Laboratory, State Key Laboratory of Agrobiotechnology, Institute of Environment, Energy and Sustainability, The Chinese University of Hong Kong, Hong Kong, China; 2The Swire Institute of Marine Science and School of Biological Sciences, The University of Hong Kong, Pokfulam Road, Hong Kong, China

**Keywords:** ecology, evolution, immune response, host–parasite interaction, invasive species, phylogeny, schistosomiasis

## Abstract

Schistosomiasis, or also generally known as bilharzia or snail fever, is a parasitic disease that is caused by trematode flatworms of the genus *Schistosoma*. It is considered by the World Health Organisation as the second most prevalent parasitic disease after malaria and affects more than 230 million people in over 70 countries. People are infected via a variety of activities ranging from agricultural, domestic, occupational to recreational activities, where the freshwater snails *Biomphalaria* release *Schistosoma* cercariae larvae that penetrate the skin of humans when exposed in water. Understanding the biology of the intermediate host snail *Biomphalaria* is thus important to reveal the potential spread of schistosomiasis. In this article, we present an overview of the latest molecular studies focused on the snail *Biomphalaria*, including its ecology, evolution, and immune response; and propose using genomics as a foundation to further understand and control this disease vector and thus the transmission of schistosomiasis.

## 1. Introduction

Schistosomiasis is a tropical parasite-borne disease resulting from the infection of trematode blood flukes of the genus *Schistosoma* [1]. It is the second most prevalent parasitic disease after malaria. According to the World Health Organization, more than 230 million people were infected globally by schistosomiasis in 2019, and approximately 700 million people currently reside in 78 countries with a high risk of infection [2]. In Africa alone there is an annual death of 280,000 people from schistosomiasis [3,4].

### 1.1. Lifecycle of Schistosoma mansoni

To understand how this parasitic disease can become so successful in spreading amongst humans, it is necessary to understand its life cycle associated with the intermediate snail host and the definitive mammalian host (mainly humans, Figure 1). In brief, fertilized eggs are shed and released via the feces or urine of infected human hosts which further develop into ciliated miracidia in freshwater [3,5,6,7]. In the presence of *Biomphalaria*, miracidia will penetrate and infect this intermediate host [3]. Inside the snail, miracidia undergo asexual reproduction to produce sporocysts [3,5,6,7], which develop to form the motile cercariae with bifurcated tails and penetration glands [6,7]. Cercariae are released from the intermediate host approximately one month after the infection [3], and a single snail can shed up to 600 cercariae per day [8]. Humans are infected by cercariae via contact with contaminated water where the cercariae penetrate the human skin, causing “swimming itch” [5,9]. Inside the human body, cercariae lose their bifurcated tail and transform into schistosomula, which enter the blood circulatory system and migrate to the liver [3,7,9]. Schistosomula grow and mature into adult worms in the portal venous system four to six weeks after infection [3,5]. The adult worms pair with another individual with a opposite sex, then migrate into the bladder or intestines for mating and egg production [7,10]. Fertilized eggs are excreted after staying in the human body for around a week to start a new cycle [3] (Figure 1).

### 1.2. Symptoms of Schistosomiasis

For infected persons, acute schistosomiasis or Katayama fever may occur due to hypersensitivity to schistosomula in the blood vessels [11,12]. Excess antigen-antibody complexes lead to various non-specific allergic symptoms such as fever, headache, myalgia, fatigue, and urticarial rash [3,5,7,12]. Most infected people recover and show no symptoms within three months [12]. Some patients, however, develop more severe clinical manifestations, such as shortness of breath, diarrhea, and enlarged liver and spleen [3,12]. Parasitism by *Schistosoma* can also cause chronic infection. The proteolytic enzyme that is secreted by the entrapped parasitic eggs stimulates granulomatous reactions and chronic eosinophilic inflammation [13,14]. The formation of granuloma in the liver results in severe abdominal pain and hematochezia [15]. Continuous liver inflammation leads to tissue fibrosis, causing portal vein thrombosis and pressure elevation [3,16] which disrupt the tissue and function of the infected liver [16].

### 1.3. Prevention and Control of Schistosomiasis

The World Health Organization (WHO) suggests the use of praziquantel for preventive chemotherapy [17], however this approach has been considered to have a limited effect on disease transmission management including the reports of praziquantel resistance in *Schistosoma* [18,19]. Given the fact that the exposure risk of schistosomiasis is highly determined by the population size of its intermediate host *Biomphalaria*, the development of a strategy to block the transmission of schistosomes by controlling the population of the snail is a strong control candidate [20]. Investigations of the efficiency of molluscicidal compounds in eliminating the snail host have been conducted for more than half a century [21]. Among the chemical compounds that have been tested, niclosamide was previously identified by WHO as the most commonly used [22], which restricts the production of ATP by the uncoupling of oxidative phosphorylation [23]. Despite a reduction in the prevalence of schistosomiasis in Brazil [24], the application of niclosamide has declined since 1986 due to the discovery of genotoxic and carcinogenic effects to non-targeted species and humans [25]. Aside from chemical molluscicides, recent research has focused on molluscicides extracted from medicinal plants. Earlier studies mentioned that plants from the families Euphorbiaceae (e.g., *Euphorbia royleana*) and Phytolaccaceae (e.g., *Phytolacca dodecandra*) have the most potent ability to kill snails [26,27]. The latex that is produced by Euphorbian is considered the most effective molluscicidal agent, which causes anaphylaxis and inflammatory responses in snails due to the effects of low pH, resulting in cell death and organ dysfunction [28,29]. Nevertheless, Euphorbian latex is a non-selective molluscicide that eliminates not only the intermediate snail host of *Schistosoma* but also other non-target aquatic species [30,31]. Recently, linalool (the extracts of *Cinnamomum camphora*) was found to cause gill damage and hepatopancreas shrinkage in snails with no observed adverse ecological impacts, thus linalool is considered as a potential powerful molluscicidal agent for *Biomphalaria* management in the future [32].

## 2. Biogeography and Evolutionary Trends of Snails *Biomphalaria*

Freshwater snails in the genus *Biomphalaria* are the intermediate host of the parasitic blood fluke *Schistosoma mansoni* and have a broad geographic distribution globally. To date, a total of 34 *Biomphalaria* species have been identified [33,34], 18 of which are potential intermediate hosts of *Schistosoma mansoni* [35]. Among the 18 *Biomphalaria* species which may act as potential intermediate host of *S. mansoni*, all 22 African species are susceptible to this parasite, but only 6 out of 12 neotropical species were tested to be infested naturally or experimentally [36,37,38,39]. To understand the spread of schistosomiasis, we, therefore, need to know how the different species of *Biomphalaria* snails have evolved. To date, different hypotheses of the evolution of species in the genera have been postulated.

### 2.1. Origin and Diversification of Biomphalaria

In the Gondwanaland origin hypothesis, based on fossil records, the last common ancestor of *Biomphalaria* was suggested to have originated and undergone speciation in Gondwanaland before the breakup of landmasses into today’s South America, Africa, Antarctica, and Australia at 100 million years ago (mya), while two major *Biomphalaria* lineages further split and separated on both sides of the Atlantic Ocean after the breakup of the supercontinent [40,41,42,43]. This hypothesis is not, however, supported by the degree of genetic differences between *Biomphalaria* species. If this hypothesis was correct, the genetic distances between *Biomphalaria* species within the same biogeographic realm should be in a range of 0.20 to 0.60 [44,45]. Woodruff and Mulvey [46], however, found that the American *B. glabrata* was distinctly separated from other Neotropical *Biomphalaria* species (mean D = 0.68) and clustered with the African species (mean D = 0.43), which implies that the formation of two lineages in South America and Africa is a more recent event than previously thought. In addition, the oldest fossil of *Biomphalaria* found in Africa was in the late Pleistocene strata (1–2 mya) [47], while those that have been found in South America occurred in the Paleocene (55–65 mya) [48,49], indicating the *Biomphalaria* that is found in Africa evolved at a later time. Similar results were also obtained from molecular analysis using mitochondrial COI gene, 16S rDNA, ITS1, and sequences, where the American *Biomphalaria* appear to be the basal taxa of their African congeners [50,51], and it is estimated that a *B. glabrata*-like species invaded Africa from 2.3 to 4.5 mya [46,49].

In the America origin hypothesis (Figure 2), a *B. glabrata*-like species experienced a trans-Atlantic radiation from South America to Africa either by attaching to the feathers of waterfowl or rafting on vegetation as egg masses, planktonic larvae, or even adults [46,51]. During the Quaternary, fluctuation in the amount of solar radiation received, repeated drought, and flooding all occurred, which resulted in periodic glaciations, and the subsequent fragmentation and coalescence of habitats [52,53]. The changes in landform and terrain facilitated the genetic divergence and speciation among populations of *Biomphalaria* [50,51], with *B. glabrata*-like species invading West Africa and evolving into other African species during the Pliocene in approximately 1.8–4.5 mya [46,50,51]. The *B. glabrata*-like species diverged into *B. camerunensis* and *B. pfeifferi* in West Africa after its colonization [51,54], with *B. pfeifferi* expanding the population range during the warm interglacial period from 30–70 kya after the last ice age [54,55,56,57]. Based on molecular phylogenetic trees, *B. pfeifferi*-like species further diverged into *B. angulosa* in East Africa, which acts as the basal member of the Nilotic species complex (*B. alexandrina*, *B. choanomphala*, *B. smithi*, *B. stanleyi*, and *B. sudanica*) [51,54].

### 2.2. Origin and Diversification of Schistosoma

In the Asian origin hypothesis (Figure 3), the common ancestor of *Schistosoma* arose in Asia and expanded to Africa [58], and this is well supported by molecular phylogenetic analyses where the Asian species (except *S. indicum*) forms a basal clade to the African congeners [58,59,60]. Using the ITS2 sequences, the separation of African species from the Asian ancestor was estimated to occur at around 24–70 mya, while the speciation of *S. mansoni* happened between 10 and 30 mya [61], suggesting that *Schistosoma* colonized Africa much earlier than *Biomphalaria*. The long-term coexistence of *Schistosoma* and *Biomphalaria* in Africa resulted in host-parasite coevolution. It has been suggested that the schistosome susceptible *B. glabrata*-like species in Africa developed parasitic resistance genes under selective pressure, causing natural selection to favor *S. mansoni* with high infectivity, such that all *Biomphalaria* in Africa are susceptible to infection [51,62,63]. On the other hand, during the 16th to 19th centuries, *S. mansoni* was introduced from the Old World to the Americas through the Atlantic slave trade as suggested by the low electrophoretic variation in enzymes between these populations [64,65]. Due to the relatively recent introduction of the parasite, it has also been suggested that no coevolutionary relationship has been established for *Schistosoma* and *Biomphalaria* in the Americas [51], and only three *Biomphalaria* species are now the natural host of *S. mansoni* in the New World [35,66,67].

## 3. Distribution and Population of *Biomphalaria* in Asia

In the Neotropical region, *Biomphalaria straminea* is the most important intermediate host of *Schistosoma mansoni* [68]. Originally native to the freshwaters of northeast Brazil [69,70], the population has now spread to the Caribbean [71,72] and other South American countries, including Paraguay, Argentina [73], and Uruguay [74] due their high desiccation tolerance and reproductive potential [75,76].

Apart from this regional expansion, transoceanic dispersal of *B. straminea* has also been reported (i.e., from Latin America to Asia). In 1973, *B. straminea* was first discovered in the Lam Tsuen River of Hong Kong [77]. This introduction has been suggested to be linked with aquarium plants and ornamental fish trade [70,78,79], which is supported by the fact that the *B. straminea* in Hong Kong groups into the Brazilian *B. straminea* clade in the molecular phylogenetic tree [80]. The original population of *B. straminea* was later expanded to various agricultural areas in the New Territories, namely Shui Wai, Shek Kong, Sha Kok Mei, and Pak Shek Au, probably due to the interconnected irrigation field ditch system (Figure 4) [81,82,83]. Allozyme frequencies at polymorphic loci Aat-1, Est-1, and Est-2 suggested that the *B. straminea* population in Hong Kong was caused by multiple introduction events [81,83]. A few decades after its initial invasion, *B. straminea* seems to have established and can be found in a variety of different places in the New Territories of Hong Kong [84].

Across the border in Shenzhen (Guangdong Province, China), the presence of *Biomphalaria* was reported in 1981 [86,87]. In the molecular phylogenetic analyses, some populations (Yantian and Liu Xian Dadao) clustered with *B. straminea* from Brazil, while some other populations (Guanlan, Kui Yong, Lianhuashan Park, Shenzhen Reservoir, and Yongzhen) formed a clade with *B. kuhniana* [80]. Due to the close trading relationship between China and Brazil [88], Attwood et al. [80] proposed that the Yantian population was transported from the Port of Belem (Northeast of Brazil) to the Yantian International Container Terminal with cargo, while the Liu Xian Dadao population could be an expansion of the established Hong Kong population (Figure 4). To date, *B. straminea* can now be found in different places in southern China including Shenzhen, Dongguan, Huizhou, and Puning [76,87,89], and is predicted to be spreading further, including Taiwan, Southern Guangxi, Fujian and other Pearl River Delta cities such as Zhongshan, Zhuhai, Jiangmen, and Yangjiang in the future [76,90,91].

Tolerance limits to different environmental stresses have been considered an important determinant for predicting the geographic distribution of *Biomphalaria* and determining their habitat suitability [76,90]. Previous research found that regional temperatures and salinity exert a strong influence on the distribution of *Biomphalaria* [92,93,94,95,96,97]. Earlier studies on *B. pfeifferi* reported that the optimal temperature for egg production was between 19 °C and 30 °C, whereas no eggs would hatch at 35 °C, resulting in the absence of this species in the African countries that lie close to the equator [93,98,99]. Interestingly, Yipp [82] reported that *B. straminea* has the ability to survive and reproduce at temperatures up to 35 °C in Hong Kong, contributing to the successful establishment of this invasive species. In terms of salinity, previous studies on *B. arabica* showed that 100% mortality occurred at 7.2‰ [97], while *B. glabrata* has a high survival rate even at 7.7‰ [96]. The greater tolerance of *B. glabrata* to more hypersaline conditions supports the records of this species in the coastal areas of Brazil [96].

## 4. Immune System of *Biomphalaria*

### 4.1. Inducible Immune Receptors

In the innate immunity of *Biomphalaria*, pathogen-associated molecular patterns (PAMPs) of invaded parasites have been detected [100]. Parasites would first be recognized by pattern recognition receptors (PRRs), and in *Biomphalaria*, fibrinogen-related protein (FREPs) is the most well-studied PRR consisting of C-terminal fibrinogen-related (FBG) domain and one or two N-terminal immunoglobulin (IgSF) domains [101,102]. The expression of FREP2, 3, 4, and 6 was reported to be upregulated in both resistant and susceptible *Biomphalaria* following schistosome infection [103,104,105,106,107], and knockdown of FREP3 will result in one-third of the resistant *B. glabrata* becoming susceptible to *Schistosoma* infection [108]. Moreover, knockdown of FREP2, 3, and 4 resulted in approximately 15% of the primo-infected snails being infected during secondary challenges while the immune memory protected all primo-infected snails in the control group, suggesting their roles in innate immune memory [109]. Several FREPs also could combine with the thioester protein (TEP) and biomphalysin to form an immunocomplex that further interacts with the polymorphic mucins antigens of *S. mansoni* (SmPoMucs) [110,111,112,113,114]. The TEP family is known as a vital element in the phagocytosis of pathogens in insects [115], and in *Biomphalaria* activated TEP is involved in the opsonization process of pathogens and parasites, which actuates the phagocytosis of foreign cells [116,117].

Another group of PRRs is the Toll-like receptor (TLR), where the upregulation of its expression was observed in resistant snails at 12 and 24 h after schistosome invasion [118]. Previous research described that the parasitic PAMPs attach to the TLRs and activate a series of downstream signal transduction cascades, including the nuclear factor kappa B1 (NF-κB1) that further promotes the expression of defense genes for inflammatory and anti-apoptotic processes [100,105,106,119,120].

### 4.2. Cell Signaling

Various cytokine sequences were also found in the genome of *B. glabrata*, including 12 interleukins (ILs), 11 tumour necrosis factors (TNF), and 4 macrophage migration inhibitory factor (MIF) [106]. Cytokines are well-known for their roles in anti-microbial responses [121,122], and in vertebrates, cytokines bind to receptors (e.g., TNFR, IL-17R) and trigger the signaling cascades of NF-κB, mitogen-activated protein kinase (MAPK), and activator protein 1 (AP1) pathways [123]. In the NF-κB pathway, the activated receptor stimulates the enzyme complex IKK and the inhibitory protein IκBα [124], while in the MAPK pathway, the ligand-activated receptor stimulates the activation of either c-Jun N-terminal kinases (JNKs) or p38 MAPK [125,126]. The identification of these signaling components in *Biomphalaria* suggests that the snail host could also rely on them to combat parasite infection [119,120,124,126].

### 4.3. Inducible Immune Effectors

In addition to the above factors that are associated with parasite recognition and cell signaling, several molecular components related to immune effectors were also found to be differentially expressed in *Biomphalaria* after infection. One of these is the oxidative attack strategy [127,128,129]. In the presence of antigens, the NADPH oxidase complex will be activated and subsequently generate reactive oxygen species (ROS) to kill the sporocyst [127,129]. At the same time, *Biomphalaria* have also developed antioxidant molecules such as Cu/Zn superoxidase dismutase (SOD) which act to avoid the damaging effects of ROS on its own cells. SOD can convert toxic superoxide radicals into oxygen (O_2_) and hydrogen peroxide (H_2_O_2_) molecules [130], and upregulation of Cu/Zn SOD genes are observed in response to the penetration of *S. mansoni* and other infectious agents [103,104,107,130,131]. Other enzymes which can break down hydrogen peroxide (H_2_O_2_) into water were also found to have their expression upregulated upon schistosome infection [132], including peroxidase, glutathione peroxidase, peroxinectin, and dual oxidase [119,133].

Last but not least, heat shock proteins (HSPs) are another notable family of proteins that are upregulated to manage stressful conditions after schistosome infection in *Biomphalaria*. In innate immunity, HSPs act similarly to antigens and bind to the immune receptors and activate signaling cascades to facilitate antimicrobial responses [134]. Upon infection, HSPs act as molecular chaperones to support the folding and conformation of proteins that are associated with the inflammatory process [135,136]. Th expression of HSP70 was upregulated in the juveniles of both resistant and susceptible snails upon schistosome penetration, but in the adult stages, upregulation of HSP70 expression was only detected in schistosome-resistant snails [137,138].

### 4.4. Cellular and Humoral Effectors

Similar to other invertebrates, the immune system of *Biomphalaria* is comprised of both cellular and humoral responses [139]. Hemocytes are the circulating cells that are involved in the invertebrate cellular defense by encapsulation and phagocytosis of sporocysts and other pathogens [140,141,142]. Previous histological studies have indicated that the migration of hemocytes to the infiltration site is quicker in resistant snail species than in species with higher susceptibility to schistostomes [139,143,144]. Thus, hemocytes have been considered as one of the main effectors of *Biomphalaria* in antiparasitic responses. In addition, injection of cell-free hemolymph from a resistant snail to a susceptible snail strain significantly reduced schistosome infective rates, suggesting soluble proteins in the hemolymph also play an indispensable role in the immune system [145,146].

### 4.5. Compatibility Polymorphism between Biomphalaria and Schistosoma

Within a population, it appears that only some individual *Biomphalaria* snail hosts are successfully infested by the schistosome, while others are incompatible [147]. The reasons for this phenomenon, however, are not yet wholly understood. There are two primary hypotheses that have been suggested, including the “resistance hypothesis” [63] and the “matching hypothesis” [148]. For the resistance hypothesis, Webster and Davies [63] proposed that the resistance and susceptibility status of the snail hosts are the main determinants for successful infection. An experiment that was conducted by Allan et al. [149] showed that there is a low cercarial shedding rate (as an indication of infection of the snail) even when exposing the snail host to an environment with high schistosome density, suggesting that resistance status rather than the encounter rate plays an important role in the lack of successful infection. In support of this hypothesis, previous research has found significant expression differences in immune-related genes between compatible and incompatible snails, such as the putative immune receptor FREP3 [108], Cu/Zn superoxide dismutase (SOD) [130,131], and growth factor granulin (BgGRN) [150]. Vulnerable snails, therefore, usually lack the ability to recognize the parasite or produce effective effector cells [147].

On the other hand, the matching hypothesis suggests that infection success is due to the matched phenotype composition between the infected snail and the schistosome [148]. Theron et al. [151] revealed that all snail hosts are potentially susceptible to infection. Successful infection occurs when increasing the population of miracidia and raising the phenotypic diversity accordingly, in order to enhance the probability of finding a matched phenotype [147,151]. In experimental treatments, the infection rate of snails decreased by more than half after transferring the experiment from the field to the laboratory. This was interpreted to be a result of the presence of a reduced set of phenotypes in the small miracidia population due to genetic drift in the laboratory as compared to the field situation, supporting the hypothesis that successful infection increases with the phenotypic diversity of miracidia [151,152,153,154].

## 5. Genomics of *Biomphalaria*—A New Angle to Shed Light on Traditional Knowledge?

The control of schistosomiasis has traditionally relied on therapeutics, but increasing levels of resistance to existing drugs has decreased the efficacy of these treatments. The advancement and feasibility of sequencing technologies can now, however, offer better opportunities to understand the intermediate host *Biomphalaria* from the genome-wide perspective. The genomic data provide information on the introduction routes and population movement of invasive species via monitoring the changes of single nucleotide polymorphisms (SNPs) among the populations. For instance, the study of population genomic structure on *Aedes albopictus*, an important vector of numerous viral pathogens, discovered that the population in Italy was caused by multiple recent colonization events [155]. The genetic mixture of various populations generated novel genotypes with higher stress adaptation in *Ae. albopictus* and resulted in rapid range expansion [155]. This finding exposed the importance of monitoring the new introduction to high-infested areas, in order to prevent the importation of resistant populations. In addition, whole genome sequences contain the sum of all genetic information that can uncover the association between the genotypes and the expressed phenotypes. Analyzing the information on genome allows us to identify novel genes and gene functions which is essential in ecological responses and interactions. As such, it should be possible to develop specific pesticides by altering crucial metabolic or infection pathways. Faucon et al. [156] identified the genes that are associated with insecticide resistance in *Ae. albopictus* from the genome, which allowed the scientists to track down the metabolic resistance pathways and develop a new insecticide that was specified for resistance mosquitoes. Special interest is also given to the infection resistance and susceptibility of snail hosts. The identification of novel candidate resistance genes enables us to resolve the problem of why only some snail hosts are compatible with the schistosomes. A better understanding of the genomes of *Biomphalaria* will allows us to recognize unique characteristics in resistant and susceptible snails, which can be selected as new targets for molluscicides that target only the susceptible snails, avoiding detrimental effects on the native ecosystem. Furthermore, modification of the targeted sequence using CRISPR technology can potentially make susceptible snails infection-resistant, reducing or even blocking the spread of schistosomes. For example, Peng et al. [157] identified that the CsLOB1 gene promoter contributes to the susceptibility of orange *Citrus sinensis* against the bacterium *Xanthomonas citri* that causes lesions on fruits and leaves. Modification of the CsLOB1 gene promoter using the CRISPR/Cas9 technology-generated transgenic plants that are resistant to the disease citrus canker [157].

To date, the genomes of *B. glabrata* [106] and *B. straminea* [158], are publicly available. The genome assembly of *B. glabrata* is approximately 916 Mb in size with a scaffold N50 length of around 48 kb. This important genomic study that was conducted by Adema et al. [106] identified the immune and stress responses of *Biomphalaria*, including Toll-like receptors (TLRs), fibrinogen-related proteins (FRPs), and heat-shock proteins (HSPs) [106]. In addition, genes that are involved in the cytokine signaling pathway (i.e., IL17, MIF, TNF) and apoptotic pathway (i.e., BIR) were discovered for the first time in *Biomphalaria* which could be potential new targets for designing specific molluscicides [106]. Other than that, this study also demonstrated the expression of core cardiac genes in *Biomphalaria* heart tissues, and the independent divergence of actin genes in molluscs, which provides a better understanding of how *Biomphalaria* evolves [106].

Recently, the genome of *B. straminea,* which is now the dominant species spreading schistosomiasis in Asia, has also been obtained and analyzed [158]. The genome assembly size of this species is around 1Gb which is similar to *B. glabrata*; and yet, it has a scaffold N50 length of more than 25 Mbp [158]. Similar to another study, this recent study also revealed new biological insights, including the identification of the first set of ecdysteroid biosynthetic pathway genes, genes that are involved in cholesterol metabolism, as well as insect sesquitperneoid pathway genes in *Biomphalaria straminea* [158]. In addition, the authors revealed a sesquiterpenoid hormone responsive system in *Biomphalaria* [158], which could also provide potential new targets for making specific molluscicides, given the fact that sesquiterpenoid juvenile hormone has been an effective target for insecticides.

The key remaining question is what else can these genomic resources reveal? Here, we argue that at least two other potential directions in addition to those above, should be explored. First, in both genomes, HSP20, HSP40, HSP60, HSP70, and HSP90, which function in mediating biotic and abiotic stressors have been identified [106,158]. Understanding the effect of different stresses on the HSP family may help us to understand how *Biomphalaria* species adapt to their environment. Another direction will be carrying out population genomics to reveal the relatedness and connectivity of individuals collected from different places, which will help us to better understand the invasion route and spread pattern of the *Biomphalaria* population, in order to predict the potential distribution in the future. Recognizing areas with high invasion risk allows the government to formulate specific pathway-based preventive strategies.

## Figures and Tables

**Figure 1 ijms-24-04895-f001:**
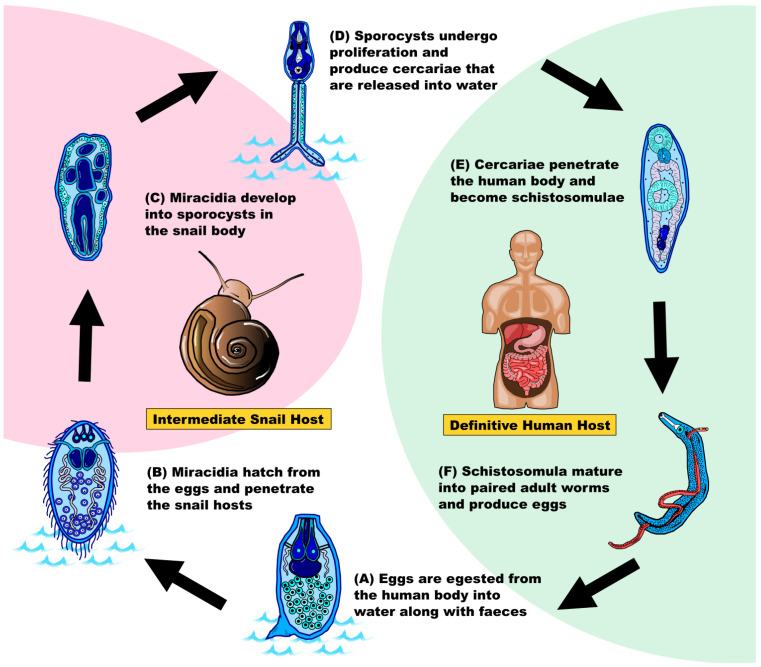
**Schematic diagram showing the lifecycle of *Schistosoma mansoni*** (after [3,5,6,7]). Pink shading indicates the lifecycle in the intermediate snail host *Biomphalaria*, white shading indicates the lifecycle in the aquatic environment, and green shading indicates the lifecycle in the definitive human host. (**A**) Egg. (**B**) Ciliated miracidium. (**C**) Sporocyst (mother sporocyst and daughter sporocyst). (**D**) Free-living cercariae. (**E**) Schistosomulae. (**F**) Paired adult worms (male: blue; female: red).

**Figure 2 ijms-24-04895-f002:**
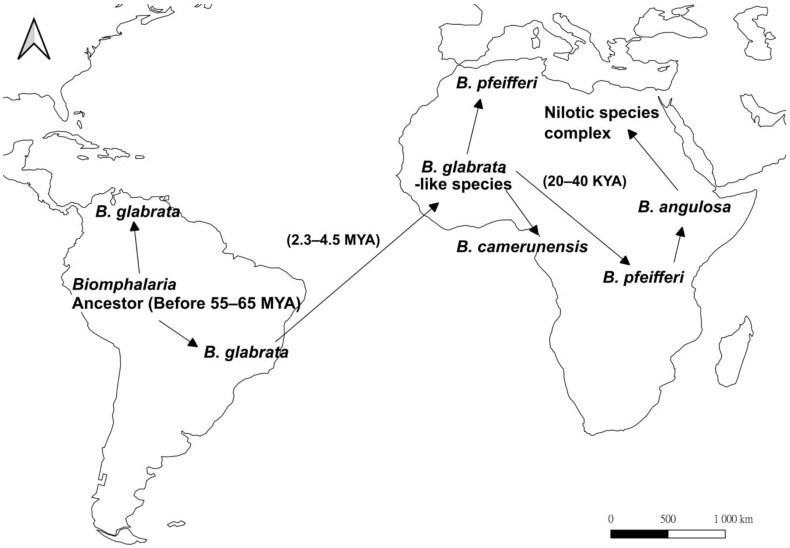
Map showing the proposed evolutionary history of the snail genus, *Biomphalaria* based on the American origin hypothesis (after [46,51]). Arrows represent proposed dispersal direction. The ancestor of *Biomphalaria* existed in South America from 55 to 65 million years ago (mya), and later diverged into more than 20 species. *B. glabrata*-like species underwent trans-Atlantic dispersal in 2.3–4.5 mya, separating into *B. camerunensis* and *B. pfeifferi* in West Africa. *B. pfeifferi* colonized East Africa in approximately 20–40 kya, which further diverged into *B. angulosa* and subsequently the Nilotic species complex.

**Figure 3 ijms-24-04895-f003:**
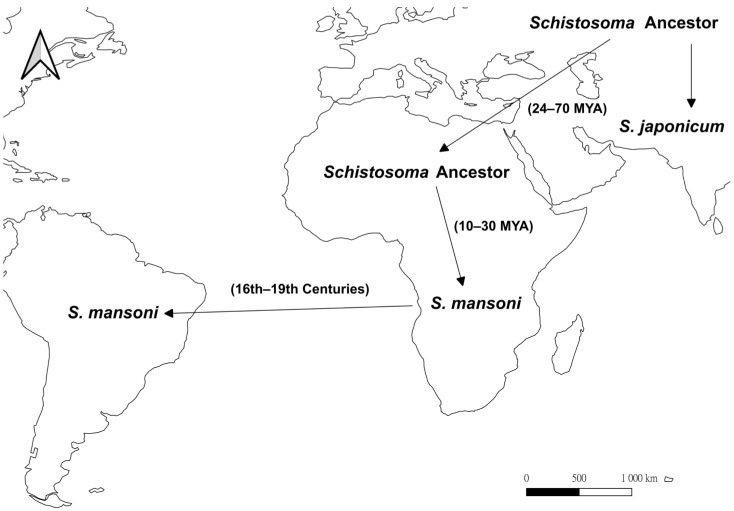
Map showing the proposed evolutionary history of the blood fluke, *Schistosoma* based on the Asian origin hypothesis (after [58,59,60,61]). Arrows represent the suggested dispersal directions. In this hypothesis, the ancestor of *Schistosoma* originated in Asia, migrated to Africa at 24 to 70 mya, and separated into *S. mansoni* and other *Schistosoma* species at 10 to 30 mya. Later in the 16th to 19th centuries, *S. mansoni* was accidentally introduced into South America through human activities such as slave trade.

**Figure 4 ijms-24-04895-f004:**
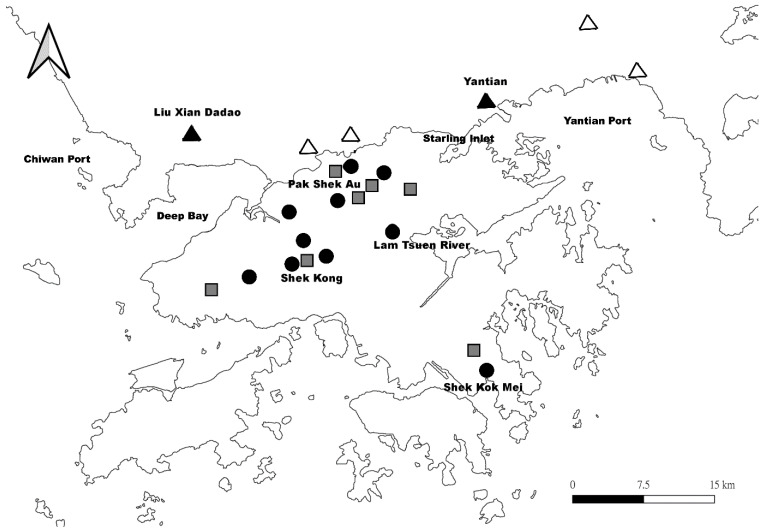
**The geographic distribution of the snail, *Biomphalaria* in Hong Kong and Southern China.** Black circles represent the distribution of *B. straminea* reported in Yipp [85]. Grey squares represent the distribution of *B. straminea* reported in Zeng et al. [84]. Black and white triangles represent the distribution of *B. straminea* and *B. kuhniana* reported in Attwood et al. [80], respectively.

## Data Availability

This study does not report any data.

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
