# Peer review of "Status Quo and Future Perspectives of Molecular and Genomic Studies on the Genus *Biomphalaria*—The Intermediate Snail Host of *Schistosoma mansoni"

_ijms, 2023, doi:10.3390/ijms24054895_

Round 1
Reviewer 1 Report
The paper submitted is a review paper. The title of the work suggests that the article will be about the disease (schistosomiasis) - but this is not the case.
This is a very cursory, generic approach to the subject. The paper adds nothing new to the problem addressed; nor is it a critical analysis of the available data on schistosomiasis, including Biomphalaria hosts.
Therefore, I do not recommend this paper for publication in the International Journal of Molecular Sciences.
Author Response
Thank you very much for pointing this out. We apologise that the title of this manuscript may be a little bit misleading. The main focus of this review is the intermediate snail host Biomphalaria rather than the disease schistosomiasis. Therefore, we have changed the title of this manuscript to “Status quo and future perspectives of molecular and genomic studies on the genus Biomphalaria – the intermediate snail host of Schistosoma mansoni”, in order to make it more clear and avoid any ambiguity (Lines 1-3).
The results of whole genome sequencing of Biomphalaria glabrata and B. straminea have made a significant change to the molecular studies of these snail hosts. The genomic data greatly improves our understanding on the biology of Biomphalaria, especially the immune system and the snail-parasite interactions. Our manuscript is the first study summarising the findings from the recently sequenced Biomphalaria genomes. We provide insight into the application of genomic resources to understand and manage the intermediate host Biomphalaria, including the detection of invasion routes, the production of species-specific molluscicides, and the generation of transgenic snails to prevent infection.
Additionally, our manuscript provides a comprehensive review of the distribution and population of Biomphalaria in Asia. Although this invasive snail has been introduced to Asia for around half a century, a summary of the distribution is lacking. This information allows us to understand the population expansion patterns and predict the future distribution of Biomphalaria in Asia, which helps us to develop effective strategies for blocking the importation and removing the existing populations, in order to avoid the potential risks of schistosomiasis transmission in Asia.
As such we believe the MS does offer some new, and indeed novel insights into the topic and, using the sequencing data, provides a new lens through which to view the issues regarding the potential means of controlling the transmission of schistosomiasis via the host vector.
Reviewer 2 Report
This is a great review article. It is succint, concise and clear in its explanation of schistosomiasis. The English is well written and the article flows.
There are however a few typographical errors that require your attention namely:
p. 2, line 78, remove commas from we, therefore, it should read, we therefore....
p. 2, lines 118-134 - Change font size appropriately
p.2, line 132 'America' should this be 'Americas' or USA?
For the strategies of intervention for the parasite, the fact that it is dessication resistant is there a way to target those genes to reduce this dessication resistance? This perhaps could offer a viable strategy for environmental control?
Author Response
[Comment 1] p. 2, line 78, remove commas from we, therefore, it should read, we therefore....
[Response 1] Thank you very much for your kind reminder. We have deleted the commas. The sentence is changed to “To understand the spread of schistosomiasis, we therefore need to know how the different species of Biomphalaria snails have evolved.” (Line 114).
[Comment 2] p. 2, lines 118-134 - Change font size appropriately
[Response 2] Thank you for your reminder. We have changed the font type from “Calibri” to “Times New Roman”, which is consistent with other paragraphs (Lines 155-158).
[Comment 3] p.2, line 132 'America' should this be 'Americas' or USA?
[Response 3] Thanks again, we have replaced the word “America” with “the Americas”. The sentence is changed to “On the other hand, during the 16th to 19th centuries, S. mansoni was introduced from the Old World to the Americas through the Atlantic slave trade as suggested by the low electrophoretic variation in enzymes between these populations. Due to the relatively recent introduction of the parasite, it has also been suggested that no coevolutionary relationship has been established for Schistosoma and Biomphalaria in the Americas” (Lines 170, 173).
[Comment 4] For the strategies of intervention for the parasite, the fact that it is dessication resistant is there a way to target those genes to reduce this dessication resistance? This perhaps could offer a viable strategy for environmental control.
[Response 4] Thank you for the suggestion. This idea provides a great new insight into the issue of schistosomiasis. The genome of Schistosoma mansoni has been sequenced by Berriman et al. (2009) and it is therefore possible to annotate the genes responsible for dessication resistance from the reference genome. Modification of the targeted sequences may produce dessication susceptible schistosomes that have a higher mortality rate in unfavourable conditions. Unfortunately, due to the main focus of this manuscript being the intermediate snail host Biomphalaria we have not elaborated on this, but it is an area worthy of future consideration.
Reviewer 3 Report
The paragraphs are too long and complex,. Readability and clarity will increase upon usage of subsections. Important relevant information are lacking.
1- In the Introduction, three aspects should be separate: 1-1. Life cycle, whereby here is the place to clarify the schistosome lifecycle in the snail rather than the final host, with a diagram more informative and useful than Figure 1, 1-2. The Disease; and 1-3. Control, whereby PZQ is mentioned while the focus in this section should rather be on molluscicides.
2-Biogeography and molecular phylogeny should belong to separate sections.
3- If you wish to have a separate section for Biomphalaria in Asia, then sections on Biomphalaria in Africa and Biomphalaria in South America should be included, and please, reflect upon their absence elsewhere despite the wide distribution of fresh water sources and more or less suitable temperatures.
4- Under immune system, please start with receptors, and under separate headings report on cellular and humoral effectors and if possible their activation mechanisms. In an entirely separate paragraph it is difficult not to tackle on the most fundamental issue of resistance to infection. An information most people are not aware of is that a negligible proportions of schistosome compatible snails in a given habitat are infected, while the majority are resistant for reasons, which remain obscure and controversial.
5- The section on genomics is confusing, and it is unclear how this area of study will 5-1. Lead to user-friendly detection procedures of infected versus intact snails, as extermination of the species is unnecessary and could be detrimental to the ecosystem. 5-2. And most importantly, prevent snail infection, the absolutely fundamental cornerstone for humane eradication of human and animal schistosomiasis.
Author Response
[Comment 1] In the Introduction, three aspects should be separate: 1-1. Life cycle whereby here is the place to clarify the schistosome lifecycle in the snail rather than the final host, with a diagram more informative and useful than Figure 1, 1-2. The Disease; and 1-3. Control, whereby PZQ is mentioned while the focus in this section should rather be on molluscicides.
[Response 1] Thank you very much for your suggestions. As proposed we have separated the “Introduction” into three subsections, including “1.1 Lifecycle of schistosome” (Line 47), “1.2 Symptoms of schistosomiasis” (Line 66), and “1.3 Prevention and control of schistosomiasis” (Line 80). Additionally, we have modified Figure 1 to make it more informative and easier to understand (Lines 411-417). We agree that molluscicides are important in Biomphalaria management. As a result we have added a paragraph discussing both chemical and phytochemical molluscicides in controlling Biomphalaria under the subsection “1.3 Prevention and control of schistosomiasis” (Lines 87-104).
[Comment 2] Biogeography and molecular phylogeny should belong to separate sections.
[Response 2] Thanks for the suggestion. Since we used phylogenetic evidence to explain the biogeographic patterns, evolutionary history and events of Biomphalaria, we find it difficult to separate “biogeography” and “phylogeny” into two independent subsections. Still, in order to make it more clear, we have separated this section into “2.1 Origin and diversification of Biomphalaria” (Line 118) and “2.2 Origin and diversification of Schistosoma” (Line 154).
[Comment 3] If you wish to have a separate section for Biomphalaria in Asia, then sections on Biomphalaria in Africa and Biomphalaria in South America should be included, and please, reflect upon their absence elsewhere despite the wide distribution of fresh water sources and more or less suitable temperatures.
[Response 3] We agree it would be wonderful to review the distribution of Biomphalaria species in Africa and South America. However in this paper, we would like to focus on the recently invaded Asian populations, which have not been reviewed before. In addition, we agree that it is important to reflect on the factors affecting the presence and absence of Biomphalaria. We have now added a part discussing how the environmental stresses (i.e. temperature and salinity) influence the spatial distribution of Biomphalaria, in order to show how it may be possible to predict the future distribution in Hong Kong and Southern China through habitat suitability (Lines 206-218).
[Comment 4] Under immune system, please start with receptors, and under separate headings report on cellular and humoral effectors and if possible their activation mechanisms. In an entirely separate paragraph it is difficult not to tackle on the most fundamental issue of resistance to infection. An information most people are not aware of is that a negligible proportions of schistosome compatible snails in a given habitat are infected, while the majority are resistant for reasons, which remain obscure and controversial.
[Response 4] Thanks for your suggestions. We have now separated the “Immune system of Biomphalaria” into five subsections, including “4.1 Inducible immune receptors” (Line 231), “4.2 Cell signaling” (Line 256), “4.3 Inducible immune effectors” (Line 269), and “4.4 Cellular and humoral effectors” (Line 292). We agree that it is important to talk about the resistance of Biomphalaria to Schistosoma, so we have added a subsection “4.5 Compatibility polymorphism between Biomphalaria and Schistosoma”, discussing two existing alternative hypotheses – the resistance hypothesis and matching hypothesis (Lines 304-329).
[Comment 5] The section on genomics is confusing, and it is unclear how this area of study will: 5-1. Lead to user-friendly detection procedures of infected versus intact snails, as extermination of the species is unnecessary and could be detrimental to the ecosystem. 5-2. And most importantly, prevent snail infection, the absolutely fundamental cornerstone for humane eradication of schistosomiasis.
[Response 5] Thank you for pointing this out as this is an important point of the paper (see reply to reviewer 1). We apologise that this section may be a little bit unclear. To address this, we have added a short paragraph with a detailed example explaining how genomic data may help us to eradicate only the susceptible snails and prevent snail infection by identifying resistant genes (Lines 351-363).
Round 2
Reviewer 1 Report
Changing the title has made the work in its current form clear and, above all, the content is consistent with the title.
Comment: chapter title - 1.1. Life cycle of schistosome:
Schistomome include various genera in the Schistosomatidae family, such as Allobilharzia, whose adult stages live only in birds, not in humans. Also, Biomphalaria snails are not always hosts for all Schistosomatidae. Therefore, it would be better and more precise to give the species Schistosoma mansoni (or only the genus „Schistosoma spp.”), instead of "schistosome," especially since the work is mainly about this species (S. mansoni) of fluke.
Author Response
Thank you very much for your kind reminder. We agree that using the species name Schistosoma mansoni in the subsection title would be more precise. We have changed the subsection title to “1.1. Lifecycle of Schistosoma mansoni” (Line 45)
Reviewer 3 Report
Great improvements to a useful review
Author Response
Thank you very much for giving valuable comments and suggestions on our manuscript.